# DisCaaS: Micro Behavior Analysis on Discussion by Camera as a Sensor

**DOI:** 10.3390/s21175719

**Published:** 2021-08-25

**Authors:** Ko Watanabe, Yusuke Soneda, Yuki Matsuda, Yugo Nakamura, Yutaka Arakawa, Andreas Dengel, Shoya Ishimaru

**Affiliations:** 1Department of Computer Science, University of Kaiserslautern & DFKI GmbH, 67663 Kaiserslautern, Germany; andreas.dengel@dfki.de (A.D.); shoya.ishimaru@dfki.de (S.I.); 2Graduate School of Science and Technology, Nara Institute of Science and Technology, Nara 630-0192, Japan; yusuke.soneda@gmail.com (Y.S.); yukimat@is.naist.jp (Y.M.); 3Department of Information Science and Technology, Graduate School and Faculty of Information Science and Electrical Engineering, Kyushu University, Fukuoka 819-0395, Japan; y-nakamura@ait.kyushu-u.ac.jp (Y.N.); arakawa@ait.kyushu-u.ac.jp (Y.A.)

**Keywords:** digital camera, camera as a smart sensor, human action recognition, meeting analysis, 3D pose estimation, RGB sensors

## Abstract

The emergence of various types of commercial cameras (compact, high resolution, high angle of view, high speed, and high dynamic range, etc.) has contributed significantly to the understanding of human activities. By taking advantage of the characteristic of a high angle of view, this paper demonstrates a system that recognizes micro-behaviors and a small group discussion with a single 360 degree camera towards quantified meeting analysis. We propose a method that recognizes *speaking* and *nodding*, which have often been overlooked in existing research, from a video stream of face images and a random forest classifier. The proposed approach was evaluated on our three datasets. In order to create the first and the second datasets, we asked participants to meet physically: 16 sets of five minutes data from 21 unique participants and seven sets of 10 min meeting data from 12 unique participants. The experimental results showed that our approach could detect *speaking* and *nodding* with a macro average f1-score of 67.9% in a 10-fold random split cross-validation and a macro average f1-score of 62.5% in a leave-one-participant-out cross-validation. By considering the increased demand for an online meeting due to the COVID-19 pandemic, we also record faces on a screen that are captured by web cameras as the third dataset and discussed the potential and challenges of applying our ideas to virtual video conferences.

## 1. Introduction

Communicating with others is one of the most important activities for coming up with new ideas, making rational decisions, and transferring skills. Many knowledge workers spend a certain amount of their work time for meetings. For instance, it is estimated that 11 million meetings are held in the workplace every day in the United States [1]. An employee’s average time spent on scheduled meetings per week is six hours, and supervisors spend 23 h [2]. The amount is increasing annually. So far we know that between the 1960s to 1980s, this has been doubled [2]. While there is no doubt about the importance of meetings, there is also another aspect in that they are time consuming for the participants and make up large costs for organizations, ranging from USD 30 million to over USD 100 million per year [3].

Regarding these facts, researchers have been investigated how to increase the efficiency and quality of meetings [4,5,6,7,8]. After a survey of publications in social science and human-computer interaction (see Section 2 for details), we found that appearances (characteristics such as age and role) [9], verbal information (e.g., spoken content/context and audio characteristics) [10,11,12,13,14,15], and non-verbal information (e.g., body gesture and facial expressions) change the behaviors of meeting participants [16,17,18,19,20]. Compared to several existing approaches for detecting/analyzing information mentioned above, our aim is to design a system that does not utilize content-sensitive information, that uses contactless devices, and that is reproducible.

Content-sensitiveness: The focus of the content-sensitiveness issue is mainly about the context relative to what was spoken in the meeting. If the transcript of the meeting leaks outside, there are high risks for the company. We adopt an approach that extracts and stores only nonverbal data from the video stream during the meeting.

Contactless: Use of contactless devices is also important for reducing time consumption of device setup. Systems must also be reproductive in order to be utilized at any location. According to these aims, we chose to collect nonverbal information for our main data. Instead of using bodily attached devices, we chose a camera as a sensor.

Reproducible: The use of one device makes the system simple, which increases reproducibility. We came up with an idea of utilizing a 360 degree camera placed at the center of the discussion table that covers all participants.

Figure 1 shows an example application that uses our micro-behavior detection method. Visualizing the timing of *speaking* and *nodding* of each meeting participant enables them to reflect upon how they were actively involved in the discussion. For instance, in the first half of the meeting, the fourth participant from the top was actively *speaking* and the second participant was agreeing by *nodding*. In the second half of the meeting, the fourth participant was agreeing with the first participant’s opinion. Therefore, our proposed work will be used for creating a system for automatic micro-behavior annotation. It will be important for both offline and online meeting analysis.

In this paper, we discuss how a camera plays an important role as a smart sensor to recognize key micro behaviors in a meeting. We propose a method that recognizes *speaking* and *nodding* from a video stream of face images and a Random Forest classifier. In order to evaluate the performance of the proposed method, we conducted data recording experiments at two physical conditions. The first recording consists of 16 sets of five minute meetings by 21 unique participants. The second recording includes seven sets of 10 min meeting data with the help of 12 unique participants. Due to the COVID-19 pandemic, most of the meetings are held online. By creating an additional dataset, we investigated whether similar features can be calculated from the web cameras connected to the PCs of each participant. In summary, we present experimental results answering the following research hypotheses:RH1: A 360 degree camera can recognize multiple participants’ micro-behavior in a small size meeting;RH2: Meetings can be recorded at any place, and the dataset can be mixed even if the collected place is different;RH3: Our camera as a sensor method can be utilized to evaluate not only offline meetings but also online meetings.

## 2. Related Work

Figure 2 shows the overview of publications investigating the importance and detection algorithms of key features in a meeting. The categories of important features can be separated into *Participants Appearance*, *Verbal Communication*, and *Nonverbal Communication*. Our target detection is not proposed in previous research studies. By showing the importance and progress of related works, we will highlight our contributions.

### 2.1. Participant Appearance

*Participant Appearance* represents features regarding the appearance of participants in meetings. Schulte et al. found that, in meetings, participants’ individual age and distribution possess a significant positive relationship with forgiveness and the amount of counteractive behaviors [9]. Forgiveness has been defined as “a reduction in negative feelings, and a recovery of positive feelings towards an offender after the offense has taken place” [21]. Counteractive statements had a negative impact on team meeting outcomes such as meeting satisfaction and team productivity [22]. Hence, the age of the meeting participant is an important feature to be considered in the meetings. Geng et al. proposed method for age estimation based on facial aging patterns [23]. They used opensource database FG-NET Aging Database [24] and the MORPH Database [25] for the dataset. These datasets include the age and face images of people. From the facial image, they extracted features by using the Appearance Model [26]. Then, by using these features, they used SVM as the algorithm for estimating age. With respect to previous works, age detection or estimation from facial image or video has already been explored [27,28].

### 2.2. Verbal Communication

*Verbal Communication* represents features regarding *speaking* that occur during meetings. McDorman states that the context of what was discussed in the meetings is important [10]. Yu and Deng have proposed a method of automatic speech recognition (ASR) technique for speech to text [29]. Shrivastava and Prasad have stated that unclear pronunciation produce unclear understanding for listeners [11]. Clear pronunciation results in high understanding of the context in meetings. In order to detect pronunciation errors, Zhang et al. have proposed a method of using deep learning technique based on advanced automatic pronunciation error detection (APED) algorithms [30]. Knowlton and Larkin have found that voice volume and pitch can enhance anxiety or provide comfort for listeners [12]. It has been also said that these factors also change the motivation of participants in terms of joining future meetings in some cases [13]. Therefore, voice volume and pitch are important.

Zhao et al. have proposed ROC Speak system, a platform that allows ubiquitous access to communication skills training. They collect voice information from the microphone in order to detect the volume and pitch of the speaker [31]. However this system has a limitation in that the user can only be collected as a single person. In the meeting situation, sometimes the voice cannot be collected due to the privacy. In order to tackle the idea of collecting verbal information concerning privacy issues, we focused on participants’ *speaking duration*. Many research studies pointed out the issue of *manterrupting* [14,15]. Essentially, this paper suggests that men speak longer than women, or men cut off the topic when a woman is talking. In order to control the equality of turns, the collection of participants speech duration is significant. We attempt to measure the duration of time by only using a camera as a sensor that does not include microphones.

Janin et al. introduced the ICSI Meeting Corpus, which collects meeting speech logs with microphones. They have used both head-mounted and table-top microphone [32]. Carletta et al. introduced the AMI Meeting Corpus, in which they used both cameras and microphones to collect meeting logs. They then created a transcript of meetings from the utterance data collected by microphones [33]. Riedhammer et al. then introduced automatic meeting summarization by using the transcript produced from ASR [34].

The advantage of verbal data analysis is the ability to collect detailed data of what was spoken in the meeting. However, the disadvantage of this approach is the privacy risk. Verbal data can be a risk during collection for a company or in the education field. It includes private or confidential information that should not be recorded. Hence, verbal data analysis will always face a privacy risk.

### 2.3. Nonverbal Communication

*Nonverbal Communication* is related to features related to actions and does not include verbal information. Posture is one of the important nonverbal information in the meetings [16,17]. Pham et al. proposed 3D pose estimation from a single RGB camera. It can estimate body posture and activities by a camera [35]. Centorrino et al. have stated that a smile perceived as honest makes mutual trust and induces cooperation towards a person who sees the smile. With this in mind [18], Zhao et al. implemented the ROC Speak system in order to collect smiles automatically from the video [31]. Bohannon et al. have stated eye contact plays a role significant for human interactions [19]. Zhang et al. then proposed eye contact detection using a camera [36]. The method can both be used by ambient cameras and wearable glasses. Kita and Ide have presented the significance of *nodding*. In this paper we focus on detecting *nodding* by a camera [20].

Ekman and Friesen stated that nonverbal information contains emotions and the feature of interaction between multiple people [37]. Effective use of nonverbal communication supports facilitating meetings and allows speakers to make attractive statements. Moreover, it is easier for listeners to understand the statement. Morency et al. used a robot to collect human head movements by camera while asking questions. They aim to collect participant’s head nods and head shakes. The recognition rate was 73% for head nods and 83% for head shakes [38]. Yu et al. used optical motion capture in order to collect nonverbal behaviors during a meeting. This approach allowed collecting *nodding* recognition of 76.4% and head shaking recognition of 80.0% [39]. A. Onishi and Terada used the IMU sensor to collect participant’s head movements [40]. They achieved the recognition of 97.5% in utterance, 52.4% in *nodding*, and 53.6% for looking around actions. The advantage of the nonverbal data analysis is to lower the risk of privacy issues. It does not contain utterance information of each meeting. Moreover, it is an effective way of understanding emotions and the features of interaction between multiple people [37].

The disadvantages of previous research studies are the inconveniences and restrictions of the system settings. Connecting IMU sensors to the participants each time before a meeting is inconvenient. The preparation of a robot inflicts restrictions in terms of the location for holding the meeting. In order to create sustainable data collection or the sustainable use of the system, it must be simple, possess fewer devices and possess low restrictions of the use case environment.

## 3. Proposed Method

This section introduces the procedures of creating a dataset and the method for feature extraction, detection, and classification.

### 3.1. Offline Meeting Data Recording

Figure 3 shows an overview of our data recording setup for offline meetings. We utilized a 360 degree camera, RICOH THETA V [41]. The frame rate was 29.97 fps, and the resolution was 3840×1920 pixels. The camera was located at the center of a circular table. The camera records all participants in the same time series. At each trial, the participant’s upper body, especially their face, must be clearly seen. We decided to record the data for a maximum of 10 min. The annotations of each action were performed by the participants.

### 3.2. Online Meeting Data Recording

We collected data of online meetings using Google Meet [42]. The frame rate was 30.00 fps, and the resolution was 1280×720 pixels. In each meeting, a range of three to four participants joined each meeting. The meeting was held for 5 min. Each time, the participant must turn on their video to show their faces. The annotation is performed by annotators.

### 3.3. Annotation of Micro-Behaviours

The annotations of micro-behaviors are performed by using ELAN [43] shown in Figure 4. ELAN is a GUI annotation tool for audio and video recordings. The users can choose to set any labels for the annotation. The participants are asked to annotate the time duration of each micro-behaviour. For our approach, *nodding* and *speaking* are within the scope of the annotation. Participants annotated right after each meeting session was completed. The annotated data are extracted as a csv file.

### 3.4. Extracting Head Rotations and Facial Points from Raw Video Frames Using OpenFace

We used the open source software OpenFace [44] to obtain features of the participant face. The images of the person after applying OpenFace and the landmarks of each facial points are shown in Figure 5. OpenFace converts video data into several features: three head rotation data ( *pose_Rx*, *pose_Ry*, and *pose_Rz* ) and 68 facial points.

### 3.5. Extracting Features from the Head Rotations and Facial Points

Following our previous work, we extracted 60 features as it is listed in Table 1 [45]. Since we aim to extract *nodding* and *speaking*, we used particular points and rotations for each micro-behaviours. For each feature, a sliding window approach is used to extract each main label in the time window. Figure 6 visualizes the process of feature extraction. We set a window frame of 1.06 Section (32 frame) with 50% (16 frame) overlap. Each annotated label is normalized as an integer by majority voting. For example, micro-behaviours such as *nodding* and *speaking* are converted into 0 and 1. For labeling, the majority of labeled numbers are selected as the main action occurring within the set time window.

*Nodding* is the action of a human individual moving its own head in the vertical direction. Hence, we focus on using the rotation feature, *pose_Rx* component shown in Figure 5. With the *pose_Rx*, we used the sliding window algorithm to extract features. We set a window frame of 1.06 Section (32 frames) and overlap at 50%. The detailed features extracted are shown in Table 1. Since we only used single features *pose_Rx*, we removed *sma*, *correlation*, and *angle*. These features are the ones calculated by using multiple features.

*Speaking* is the action of an individuals in moving their upper and lower lips. When a person speaks, the distance between the upper and lower lips becomes larger. In order to collect this feature, we used face point number 62 and 66, shown in Figure 5. The distance between number 62 and 66 is the parameter. Then, we applied the sliding window algorithm. The window frame is set at 1.06 Section (32 frames), and overlap is set at 50%. The features extracted are shown in Table 1.

### 3.6. Classification

For both offline and online meetings, we classified *speaking*, *nodding* and *other* by random forest with the calculated features for each window sample. Our preliminary experiments revealed that there is not much difference in recognition performance among machine learning algorithms. Since our approach has a large number of features, we decided to use random forest, which does not degrade recognition accuracy even with a large number of features. Since comparing the performance of machine learning algorithms was not within our main scope, we only reported results using random forests in the Evaluation Section. Hyper parameters of the following are used for classification: the number of trees, 100; criterion, Gini impurity; and the number of max features, 7 (square root of the number of features).

## 4. Experiment

In order to evaluate the performance of the proposed approach, we prepared three meeting datasets. Note that our experiments do not include any EU citizens. Therefore, the General Data Protection Regulation (GDPR) does not apply to our recordings. In this section, we explain the details of the dataset we utilized (Dataset A) and recorded (Dataset B and Online Dataset).

### 4.1. Offline Meeting Dataset A

Data were collected from 22 unique participants (18 males and 4 females) using multiple devices including 360 cameras (RICOH THETA V). Each recording was performed for five minutes. A total of 16 sets were collected. We removed the data of participants’ gaze and the acceleration data of head movements, which are included in the original dataset. The study received ethics approval (approval no: 2018-I28) after review by the research ethics committee at the Nara Institute of Science and Technology. Since this dataset is publicly available, further details are written provided in a paper by Soneda et al. [46].

### 4.2. Offline Meeting Dataset B

Data were collected from 12 unique participants (11 males and 1 female). Each recording was performed for 10 min. A total of seven sets were collected. The combination of the entire dataset becomes 34 unique participants (29 males and 5 females). The total time collected is 150 (80 + 70) min.

### 4.3. Online Meeting Dataset

For online meeting analysis, we collected data using Google Meet [42]. Our main proposal is the offline meeting analysis, but we also performed online meeting analysis for discussions in future works. The data are collected from Kyushu University, Japan. Unique participants were six people in total, and each meeting is collected for five minutes. Seventeen sessions were collected. The total time collected amounts to 85 min.

### 4.4. Evaluation Protocol

By using the model, the 10-fold random cross validation and leave-one-participant-out cross validation were applied. The 10-fold random cross-validation used a one-fold random dataset as test data and the other nine-fold random dataset as a training data. We used 10 sets perform cross-validation. In the case of leave-one-participant-out cross validation, the test data includes one participant data and train data are used for all others. Since the amount of data relative to the labeled behaviours (*nodding* and *speaking*) is lesser than non-labeled behaviour (*other*), we used downsampling methods. The data for each are reduced to balance the actions of *nodding* or *speaking*. For the machine learning technique, we used random forest for all three patterns. On the other hand, we also ran a prediction of online meeting micro-behaviour analysis. For online meetings, we classified *nodding*, *speaking*, and *other*. The 10-fold random split cross validation and leave-one-participant-out cross validation were applied.

### 4.5. Results

For offline meeting analysis, we collected data from two different places. We classified each dataset and combined them. We called them “Dataset A”, “Dataset B”, and “Dataset A + B”. As it is shown in Figure 7, we have decided to use the window size of 1.06sec (32 frame) with 50% (16 frame) overlap because the macro average F1-score was highest. The results of precision, recall, and f1-score are shown in Table 2. The confusion matrix of *nodding*, *speaking*, and *other* is shown in Figure 8. As a result, *nodding* becomes a lower f1-score than *speaking*. This result shows that *nodding* is difficult to predict compared to *speaking*. *Speaking* takes an average time of 4.01 s, and *nodding* takes an average of 1.06 s. The results of macro average f1-score for 10-fold random split cross validation are 0.69±0.05, and 0.68±0.07. Dataset B is the highest among the three dataset patterns. The results of the macro average f1-score for leave-one-participant-out cross validation are 0.62±0.09, 0.58±0.15, and 0.63±0.11. Dataset A + B was the highest among three dataset patterns. In the case of leave-one-participant-out cross validation, the lowest macro average f1-score is 0.39 and the highest is 0.78.

For the online meeting, the results of precision, recall, and f1-score are shown in Table 3. The confusion matrix of *nodding*, *speaking*, and *other* is shown in Figure 9. The result of the macro average f1-score of the 10-fold random split cross validation is 0.55±0.08. The result for the leave-one-participant-out cross validation is 0.31±0.01. In the case of the leave-one-participant-out cross validation, the lowest macro average f1-score is 0.27 and the highest is 0.35. Calculation of each function is explained in Table 1.

Regarding Table 4, the results show that, for offline and online meetings, the feature importance was different among them. The feature importance of the offline meeting is shown in Table 4a, and for the online meeting is shown in Table 4b. We have found that, for offline meetings, the majority of important features are the components related to lips. For online meetings, the majority of important features are the components related to head rotation.

## 5. Discussion

In this section, we discuss the three research hypotheses stated in Section 1.

### 5.1. Can a 360 Degree Camera Recognize Multiple Participants Micro-Behaviour in a Meeting?

Combining all datasets, the macro average f1-score is 0.68±0.07 for the 10-fold random split cross validation. The leave-one-participant-out approach macro average f1-score is 0.63±0.11. We found out that, by only using a 360 degree camera as a sensor, we could collect micro-behaviours in the meeting. Multinational classification is possible. The f1-score of *speaking* is the highest in any condition. *Speaking* recorded a higher score because the action does not vary with each participant. All participants opened their mouths while *speaking*. Therefore, we could state that using the distance of the upper and lower lip as the feature is effective. On the other hand, *nodding* seems different between each participant. When we looked at the raw video data, we discovered that some participants perform *nodding* with shallow and fast head movements, while others only perform deep and fast nodding. With respect to this, using head rotation data for predicting *nodding* scored lower than *speaking*. We also discovered that for the leave-one-participant-out approach, participants’ lowest macro average f1-score was 0.39 and the highest is 0.78. When we looked at the annotation data, we discovered that participants with the lowest f1-score had fewer labels of *nodding* than the highest participant. This is due to the difference in annotations. The participants with fewer annotations were only labeled with *nodding* that was deep and slow. This result caused less feature data for *nodding* and the score is reduced. Overall, the discussion states that *speaking* recognition from non verbal data is easier than *nodding*. In terms of the answer for *RH1*, the macro average of the f1-score achieved 0.68±0.07 and 0.63±0.11; we could say that using 360 camera as a sensor for detecting micro-behaviours is effective.

### 5.2. Can (and Should) We Extend the Dataset by Adding Data Recorded in Other Places?

In order to prove the possibility of the expanding dataset, we collected data from two different locations. The results of 10-fold random split cross validation showed that each dataset produced a f1-score of 0.65±0.05, 0.69±0.05, and 0.68±0.07, shown in Table 2. The result was the highest for Dataset B. Our assumption towards this result is the number of same participants included in the dataset. In Dataset A, 22 unique participants joined. Among them, 12 participants were involved in 20 min of the meeting, and 10 participants were involved in 10 min in total. For Dataset B, 12 unique participants joined. Among them, one participant joined for 50 min, two participants joined for 40 min, two participants joined for 30 min, two participants joined for 20 min, and three participants joined for 10 min. By comparing Datasets A and B, each participant’s duration of time joined in the meetings is larger in Dataset B. This means that the volume of individual behaviour included in Dataset B is the largest. Hence, without removing participant behaviour information as a test data, the model accuracy is the highest for Dataset B. We could refer that, by adding more personal data, the prediction rate will improve for each person when testing. With this result in mind, the leave-one-participant-out cross validation produced the f1-scores for each dataset of the following: 0.62±0.09, 0.58±0.15, and 0.63±0.11. We have found that Dataset B is the lowest in the f1-score compared to the analysis of other dataset. Once personal data are removed from the dataset, the f1-score slightly decreases. However, it is interesting that the combination of all datasets’ f1-score results in 0.63±0.11. This result is the highest compared to the prediction of the other dataset. For the answer for *RH2*, the results show that the more the dataset increases, the accuracy of the model of micro-behaviour prediction also increases.

### 5.3. Can Our “Camera as a Sensor” Method Cover Both Offline and Online Meetings?

Looking at result from Table 2, the 10-fold random split cross validation produced high f1-scores. The result of f1-score is 0.55±0.08. For the leave-one-participant-out cross validation, the score is 0.31±0.01. The classification result decreases with the leave-one-participant-out cross validation. This is probably the result of the same reason stated in the discussion of *RH1*. Individual data will be powerful for predicting certain people’s behaviour. By looking at the result of the f1-score for the individual participants, it is observed that the lowest is 0.27 and the highest is 0.35. One of the unique parts of the reduction in f1-scores in online meetings is that it is caused by the position of the face. Our proposed method for offline meeting analysis uses a 360 camera to track the entire upper body of the participants but, for online meetings, the participant face is often the only object recorded. Regarding fewer white space for face tracking, online meetings often became off track when using OpenFace [44]. For the answer relative to *RH3*, even with the concern of failure in tracking the face, the f1-score of 0.55±0.08 and 0.31±0.01 says that our camera can be potentially used as a sensor approach for online meetings.

### 5.4. Limitations

The imitations of our work include recognizing multiple actions at the same time. Our models only predict single behaviour happening in a set time range. However, the meeting behaviour is complex. *Speaking* and *nodding* can happen at the same time, but we did not consider predicting both at the same time.

Another consideration is the nervous tic of *nodding*. Some participants perform more *nodding* actions than others. However, we cannot detect whether the *nodding* performed by participants includes some context. We only detected the movement of *nodding*. Therefore, our future work includes recognition of the context inside *nodding* performed by participants.

The time duration and size of the meeting are also things we have to mention. We conducted meetings of 10 min with four people in each meeting as the largest size. In theory, our approach can be used even if we have long meetings. We only recorded meetings for 10 min maximum since we considered that the load relative to annotations for each meeting participant will be high. Our approach uses a sliding window, which means we split the video in a set amount of length so that the total time duration of meetings will not be a problem. In theory, our approach can be used even if we used meetings of a larger size if we can capture all participant’s faces clearly. For offline meetings, we have the limitation of physical space. If we want to increase the number of participants, we must add more cameras. For online meetings, the maximum number of participants displayed on a screen is limited to the device screen. Moreover, as the number of participants increases, the screen size of each participant will be smaller. Hence, for future works, we need to think about how to record the meetings with more participants.

Compared with offline and online meetings, the angle of view is one of the limitations for the online video. By using a 360 degree camera, we always tracked the participants upper body. We are able to track the participants’ face most of the time during experiment. However, for the online meeting, the angle of view is different for each participant. In particular, when participants try to observe the screen shared on their own laptops, they often move closer to the screen. In those cases, the participant’s face moves outside of the box, which results in being unable to track their face. Moreover, notifications often distract participants face. It covers the participants’ face and results in being unable to track the participant once someone types into the chat.

Since we extracted faces from screen recordings, the recording condition was not constant, i.e., our face tracking failed when the face image is hidden by a desktop notification or the face image becomes relatively smaller when someone starts sharing a screen. This problem should be solved in our future work. For instance, making our own application, bot service, or plugin for an online meeting tool are all potential directions.

For the machine learning model, we must consider the feature importance. We have found that the feature importance for the machine learning model varies with offline and online. Hence, the method is the same but we need to collect the dataset of each offline and online meeting in order to create a precise prediction model of *speaking* and *nodding*. Moreover, processing time is not considered in our work, and it will be important if we want to extend our work into real time micro-behavior recognition. Regarding these findings, the experiment setting for online meeting analysis could be further explored for future work.

## 6. Conclusions

In this work, we analyzed micro-behaviours occuring during offline and online meetings. For offline meeting analysis, we used 360 degree cameras and, for online meeting analysis, we used Google Meet. Our target micro-behaviours are (*speaking* and *nodding*). For the offline meetings, the result of the f1-score is 67.9% for 10-fold random split cross validation. For the leave-one-participant-out cross validation, it is 62.5%.We also discovered that for offline meeting data, combining the dataset collected in different places can still increase the accuracy of recognition model. From this result, we could highlight that anyone can follow our work as a framework to increase the dataset and accuracy of the model. We also applied cameras as a sensor method for the online meeting. For the 10-fold random split cross validation, we observed a macro average f1-score of 55.3%. The leave-one-participant-out approach achieved a macro average f1-score of 31.1%. We have found out that participant behaviour is important for creating an accurate model. We believe that micro-behaviour analysis using a camera as a sensor approach will become the means for new meeting analysis platforms.

## Figures and Tables

**Figure 1 sensors-21-05719-f001:**
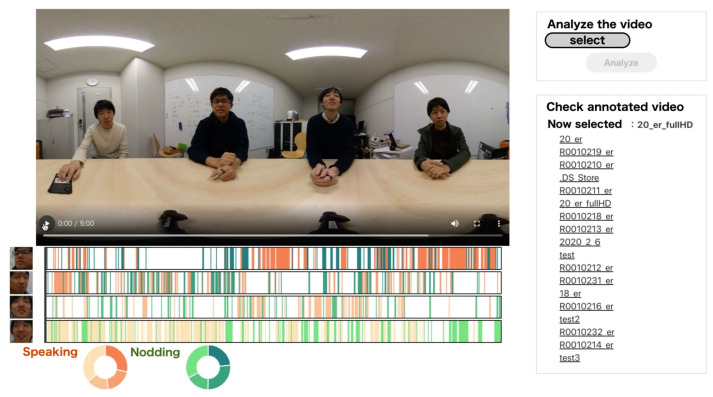
A screenshot of a meeting review system that utilizes our micro-behavior recognition. A user selects a video file, and the system classifies micro-behaviours in time series.

**Figure 2 sensors-21-05719-f002:**
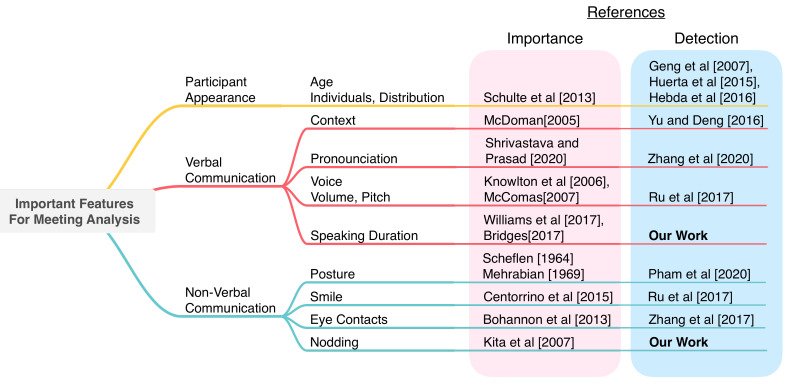
A tree map summarizing important activities for meeting analysis and the position of our work. The importance shows references representing the significance of each feature. The detection shows references of researchers implementing the system in order to detect each features [9,10,11,12,13,14,15,16,17,18,19,20,21,22,23,27,28,29,30,31,32,35,36].

**Figure 3 sensors-21-05719-f003:**
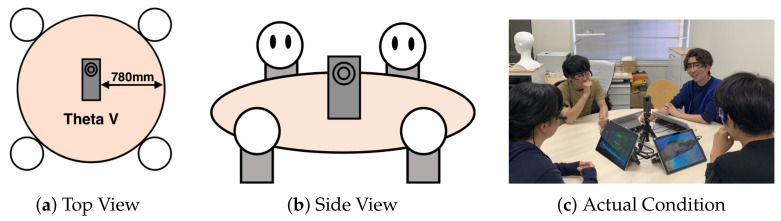
The Device Position. RICOH THETA V [41] is located at the center of the circular table. It is located approximately 780 mm from the edge of the table. (**a**) shows the view of the meeting condition from the top of the room. (**b**) shows the view of the meeting condition from the side of the room. (**c**) shows the actual scene of performing an experiment.

**Figure 4 sensors-21-05719-f004:**
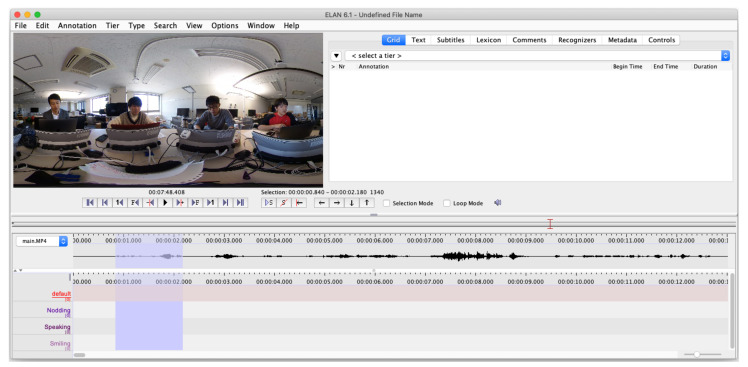
Participants use ELAN [43] to create annotations of their micro-behaviours. Each annotation contains time duration. Annotations are performed right after each meeting session was completed. After annotation, each participant exports data into a csv file.

**Figure 5 sensors-21-05719-f005:**
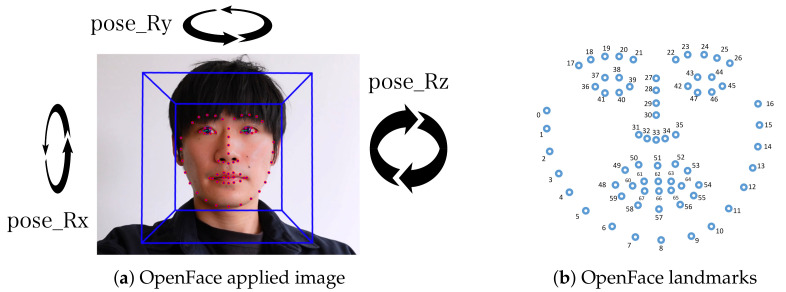
Image after applying OpenFace [44] and the 68 OpenFace landmarks of the facial points.

**Figure 6 sensors-21-05719-f006:**
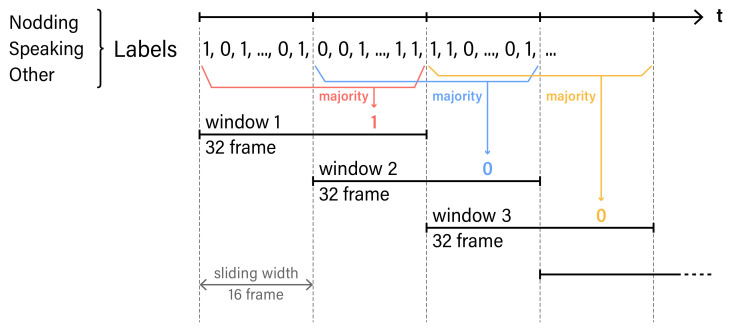
Sliding window algorithm used for feature extraction. In each time frame, there is a label of micro-behaviours. Label is normalized into an integer. One window is 32 frame. Sliding width is 16 frame. The label with the high majority will be a feature for each window.

**Figure 7 sensors-21-05719-f007:**
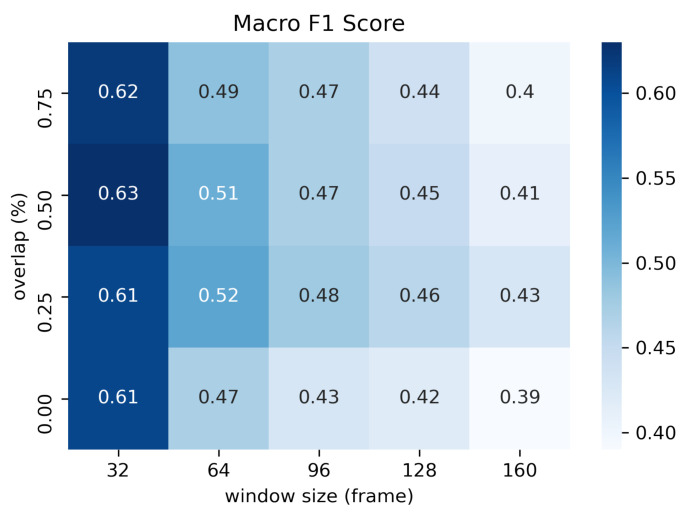
Heatmap of macro average F1-score of overlap vs. window size. All offline meeting datasets are used. The unit for window size is by frame. The unit for overlap is by percentage.

**Figure 8 sensors-21-05719-f008:**
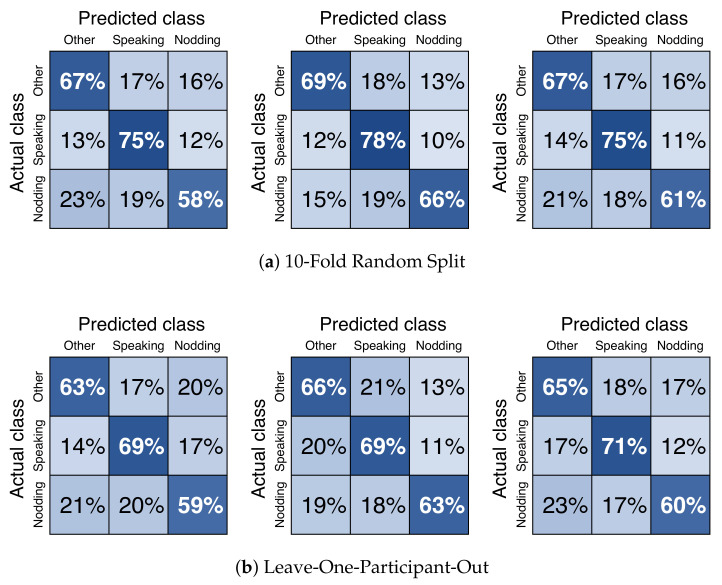
Confusion Matrix of *nodding*, *speaking*, and *other*. Using facial points and head rotation data as features. Down sampling was applied. Only offline meeting datasets were used. From the left figure, the result is extracted from M3B Corpus [46] (Dataset A), Kyushu University (Dataset B), and both (Dataset A + B). Datasets are split into two patterns: (**a**) 10-fold random split and (**b**) leave-one-participant-out. Random forest is used as the machine learning algorithm.

**Figure 9 sensors-21-05719-f009:**
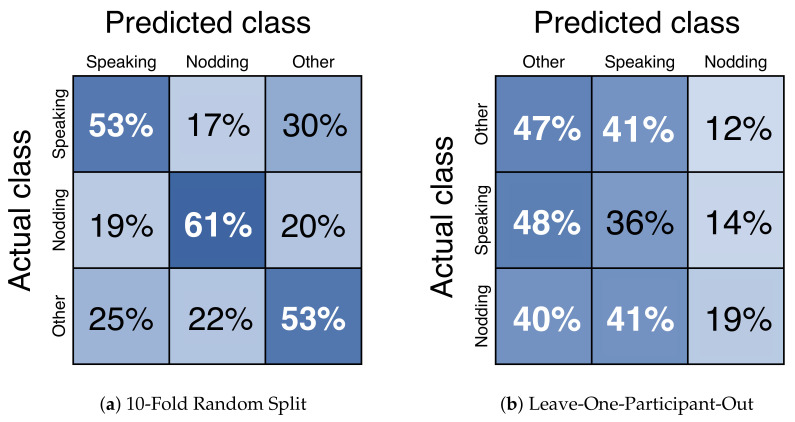
Confusion Matrix of *nodding* and *other*. Used facial points and head rotation data as features. Down sampling was applied. Only online meeting datasets were used. Dataset was split into two patterns: (**a**) 10-fold random split and (**b**) Leave-One-Participant-Out. Random forest was used as the machine learning algorithm.

**Table 1 sensors-21-05719-t001:** Feature Lists.

Function	Description	Formulation	Type
mean (s)	Arithmetic mean	$\bar{s}=\frac{1}{N}\sum_{i=1}^{N}s_{i}$	T,F
std (s)	Standard deviation	$\sigma =\sqrt{\frac{1}{N}\sum_{i=1}^{N}{\left(s_{i}-\bar{s}\right)}^{2}}$	T,F
mad (s)	Median absolute deviation	$median_{i}\left(|s_{i}-median_{j}\left(s_{j}\right)|\right)$	T,F
max (s)	Largest values in array	$max_{i}\left(s_{i}\right)$	T,F
min (s)	Smallest value in array	$min_{i}\left(s_{i}\right)$	T,F
energy (s)	Average sum of the square	$\frac{1}{N}\sum_{i=1}^{N}s_{i}^{2}$	T,F
sma ($s_{1}$,$s_{2}$,$s_{3}$)	Signal magnitude area	$\frac{1}{3}\sum_{i=1}^{3}\sum_{j=1}^{N}\left|s_{i,j}\right|$	T,F
entropy (s)	Signal Entropy	$\sum_{i=1}^{N}\left(c_{i}\log\left(c_{i}\right)\right),c_{i}=s_{i}/\sum_{j=1}^{N}s_{j}$	T,F
iqr (s)	Interquartile range	Q3(s)−Q1(s)	T,F
autorregresion (s)	Fourth order Burg Autoregression coefficients	$a=arburg\left(s,4\right),a\in R^{4}$	T
correlation ($s_{1}$,$s_{2}$)	Pearson Correlation coefficient	$C_{1,2}/\sqrt{C_{1,1}C_{2,2}},C=cov\left(s_{1},s_{2}\right)$	T
angle ($s_{1}$,$s_{2}$,$s_{3}$,v)	Angle between signal mean and vector	$\tan^{-1}\left(\right\|\left[{\bar{s}}_{1},{\bar{s}}_{2},{\bar{s}}_{3}\right]\times \upsilon \left\|,\left[{\bar{s}}_{1},{\bar{s}}_{2},{\bar{s}}_{3}\right]\cdot \upsilon \right)$	T
range (s)	Distance of the smallest and largest value	$max_{i}\left(s_{i}\right)-mix_{i}\left(s_{i}\right)$	T
rms (s)	Root square means	$\sqrt{\frac{1}{N}\left(s_{1}^{2}+s_{2}^{2}+\ldots +s_{N}^{2}\right)}$	T
skewness (s)	Frequency signal Skewness	$E[{\left(\frac{s-\bar{s}}{\sigma }\right)}^{3}]$	F
kurtosis (s)	Frequency signal Kurtosis	$E\left[{\left(s-\bar{s}\right)}^{4}\right]/E{\left[{\left(s-\bar{s}\right)}^{2}\right]}^{2}$	F
maxFreqInd (s)	Largest frequency component	$argmax_{i}\left(s_{i}\right)$	F
meanFreq (s)	Frequency signal weighted average	$\sum_{i=1}^{N}\left(is_{i}\right)/\sum_{j=1}^{N}s_{j}$	F
energyBand (s,a,b)	Spectral energy of a frequency band (a, b)	$\frac{1}{a-b+1}\sum_{i=a}^{b}s_{i}^{2}$	F
psd (s)	Power spectral density	$\frac{1}{Freq}\sum_{i=1}^{N}s_{i}^{2}$	F

N: signal vector length, Q: Quartile, T: Time domain, F: Frequency domain.

**Table 2 sensors-21-05719-t002:** Prediction Result of *Nodding* and *Speaking* for Offline Meeting.

		(a) 10-Fold Random Split		
Dataset	Label	Precision	Recall	F1-Score
A	*nodding*	0.66±0.09	0.58±0.17	0.61±0.13
	*speaking*	0.68±0.07	0.75±0.07	0.71±0.05
	macro ave.			0.65±0.05
B	*nodding*	0.73±0.04	0.62±0.15	0.66±0.10
	*speaking*	0.69±0.04	0.78±0.03	0.73±0.03
	macro ave.			0.69±0.05
A + B	*nodding*	0.68±0.10	0.61±0.17	0.64±0.14
	*speaking*	0.69±0.05	0.75±0.04	0.72±0.04
	macro ave.			0.68±0.07
		**(b) Leave-One-Participant-Out**		
Dataset	Label	Precision	Recall	F1-Score
A	*nodding*	0.64±0.17	0.60±0.17	0.60±0.14
	*speaking*	0.63±0.21	0.68±0.21	0.63±0.19
	macro ave.			0.62±0.09
B	*nodding*	0.60±0.25	0.49±0.25	0.53±0.23
	*speaking*	0.57±0.26	0.64±0.24	0.58±0.24
	macro ave.			0.58±0.15
A + B	*nodding*	0.66±0.16	0.59±0.19	0.60±0.17
	*speaking*	0.65±0.20	0.71±0.19	0.66±0.18
	macro ave.			0.63±0.11

**Table 3 sensors-21-05719-t003:** Prediction Result of *Nodding* for Online Meeting.

(a) 10-Fold Random Split			
Label	Precision	Recall	F1-Score
*nodding*	0.66±0.17	0.60±0.17	0.60±0.14
*speaking*	0.62±0.22	0.68±0.19	0.64±0.19
macro ave.			0.55±0.08
**(b) Leave-One-Participant-Out**			
Label	Precision	Recall	F1-Score
*nodding*	0.41±0.26	0.23±0.13	0.23±0.14
*speaking*	0.37±0.29	0.40±0.25	0.31±0.10
macro ave.			0.31±0.01

**Table 4 sensors-21-05719-t004:** Feature Importance of Micro-Behaviour Recognition.

		(a) Offline Meeting		
Rank	Function	Component	Type	Weight
1	iqr	distance between facial point 62 and 66	frequency	0.046
2	iqr	pose_Rx	frequency	0.040
3	std	distance between facial point 62 and 66	time	0.039
4	ARCoeff-2	distance between facial point 62 and 66	time	0.038
5	ARCoeff-1	pose_Rx	time	0.037
		**(b) Online Meeting**		
Rank	Function	Component	Type	Weight
1	entropy	pose_Rx	time	0.033
2	mean	pose_Rx	time	0.031
3	ARCoeff-3	distance between facial point 62 and 66	time	0.030
4	min	pose_Rx	time	0.029
5	Skewness-1	pose_Rx	frequency	0.027

## Data Availability

The data presented in this study are available on request from the corresponding author. The data are not publicly available due to the privacy.

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
