# Peer review of "DisCaaS: Micro Behavior Analysis on Discussion by Camera as a Sensor"

_sensors, 2021, doi:10.3390/s21175719_

Round 1
Reviewer 1 Report
The paper concerns recognition in on-line meetings three types of behaviours: speaking, nodding, and other. Authors use a set of features extracted by OpenFace toolkit and Random Forest as classifier. The paper is clearly written, however I have some objections:
- The originality of the approach questionable, the solution is typical: annotation of the data, use general purpose features extraction and good classifier. I accept the originality of the experiment.
- Authors choose arbitrary a features set and classifier, then we do not know whether it is the best or not? Most important part of the work by OpenFace was performed. At least the comparison with convolutional network would be interesting.
- Some additional information about the processing time should be added.
- In my opinion the quality is to low for at least some of practical purposes, and at least some critical discussion about the usefulness of the results should be added.
Despite the above questions my overall opinion about the paper is good - it is clearly written, it may be interesting for readers, and after some minor revision it may be accepted.
Minor remarks:
page 9,10 figure caption: "machine learning algorism."
Author Response
Dear whom reviewed our work,
Thank you so much for the review.
We have attached the cover letter.
Please check our work.

Reviewer 2 Report
The authors have investigated the possibility of using camera as sensor for the detection and classification of speaking duration and nodding during offline and online meetings. I have the following comments
- The authors should discuss the the possibility of using the proposed approach for large size and longer meetings.
- There are some typo and spelling mistakes, such as "Many" on page 4 line 109 seems to be "men", the word "honsetsy" on page 4 line 130 is not clear,
- The authors said that "We set a window frame of 1.06sec (32 frame) with 50% (16 frame) overlap." The authors should discuss the effect of the window length and overlap between consecutive windows on the classification performance.
- For the current work, the authors have used Random Forest as classifier, however, the authors did not discuss specific reason for Random Forest as classifier. In my opinion, we may never know the type of best machine learning model for a given problem without trying different models. The authors should also show the results of other types of machine learning models (SVMs, LDA, KNN, Tree classifiers, etc.).
- The meaning of the square root of the number of features as number of features is not clear on page 6 lines 206-207.
- In Figure 7, the word 'spited' seems to be misused, also, the spelling of "algorism" seems to be incorrect.
- On page 10 line 273, the authors said "Also easier to annotate makes it easier to predict" to show the reason for lower classification rate of nodding. However, in my opinion, another reason could be better discriminative features for speaking than for nodding, authors should consider and discuss the effect of features for speaking and nodding this in the revised manuscript.
- On page 11 lines 312-313, the authors said "The results slightly decrease with leave-one-participant-out cross validation." however, from 0.55 to 0.31 is not slight decrease, authors should restate the sentence.
- On page 11 lines 322-323, the authors said "Another reason can be the use of share screen." as a reason for the lower classification accuracy of speaking and nodding in online meeting. However, no share screen situation are explained for the data set of online meeting. To interpret the results and support the authors claim, screen share information should be added when introducing the Online meeting data set.
- The authors have not discussed the effect of optimum features. I recommend to discuss the effect of features, features selection, and shallow learning models of different types, in the revised manuscript.
Author Response
Dear whom reviewed our work,
Thank you so much for the review.
We have attached the cover letter and final draft of our publication.
Please check our work.
"Point-by-point response to the review by Reviewer 2 - Ko Watanabe" is the text of our comments and "SENSORS - DisCaaS - final draft.pdf" is the final draft of our work.

Reviewer 3 Report
Several suggestions for improving the manuscript.
In the Introduction sections: the several terms (content-sensitive, contactless, etc.) used and relevant for their work could be presented in a different way and also contextualize the meaning with relevant work in the field. This section could also have more work and context to help understand the novelty and the overall importance of their work. In the related work section, the type of algorithms/technology used by different authors should also be addressed. Not only mention that "X has explored that already" (e.g. in the participant appearance subsection). I would recommend writing "3D pose estimation" instead of "3d pose estimation". (line 124, page 4), and also "RGB". A reference to the "IMU sensor" should be provided. What are your thoughts and concerns regarding the General Data Protection Regulation (GDPR) and the limitations to the proposed work. Some people can have the nervous tic of nooding to much. Is the systema able to identify this feature? The way Speaking and Nodding is written should be similar in all the manuscripts. Sometimes it is in italic and others not. The authors should have replicated their study/case study with other datasets and different scenarios. The overall experiment (from preparing the datasets and obtaining the results) was tailored only to this study.Author Response
Dear whom reviewed our work,
Thank you so much for the review.
We have attached the cover letter and final draft of our publication.
Please check our work.
"Point-by-point response to the review by Reviewer 3 - Ko Watanabe" is the text of our comments and "SENSORS - DisCaaS - final draft.pdf" is the final draft of our work.

Round 2
Reviewer 2 Report
The authors have addressed all my comments in the revised version of the manuscript, I suggest the paper for publication in Sensors Journal in the current form.